# Effect of Stray Current on Corrosion and Calcium Ion Corrosion of Concrete Reinforcement

**DOI:** 10.3390/ma15207287

**Published:** 2022-10-18

**Authors:** Weijun Yang, Xin Ye, Rongjun Li, Jianyu Yang

**Affiliations:** 1School of Civil Engineering, Changsha University of Science & Technology, Changsha 410114, China; 2General Contracting Company of CCSED, Changsha 410114, China

**Keywords:** stray current, water environment, corrosion of steel bars, calcium ion erosion, compressive strength damage

## Abstract

The construction of subways, hydroelectric stations and water substations is exposed to stray currents, which can accelerate concrete corrosion. The influence of stray currents on reinforced concrete structures is unclear. In this paper, the influence of concrete strength grade, reinforcement diameter and stray current intensity on the extent of reinforcement corrosion and calcium ion dissolution were investigated, and the damage of reinforcement and calcium ion corrosion to concrete strength was investigated by simulating a stray current environment and conducting an electrified acceleration test. The test results show that the higher the strength grade of concrete, the lower the stray current intensity and the smaller the corrosion rate and calcium ion dissolution rate of reinforcement; with the increase in the diameter of reinforcement the corrosion rate decreases, but the calcium ion dissolution rate is not affected by reinforcement diameter. The damage effect of reinforcement corrosion on concrete compressive strength is more obvious than that of calcium ion corrosion.

## 1. Introduction

The metros in Chinese cities use direct current traction power supply systems. However, the insulation between the walking track and the metro bed is not completely sealed, and there will be part of the direct current based from the walking track leakage [1]. In the power stations and substations, there will be part of the current leakage from the circuit due to high-voltage transmission lines and a variety of power equipment grounding and leakage. All of the above lead to the generation of stray currents [2].

The concrete structures of subways, hydropower stations and water substations often work in water-saturated environments and suffer from the corrosion of stray currents [3]. The concrete structures being in a water-saturated environment will cause the concentration of liquid phase lime in the cement-based materials to decrease, the solid phase lime to dissolve, the cement hydration products to decalcify, transform or decompose, resulting in an increase in concrete porosity and the corrosion of the cement-based materials [4]. When the reinforced concrete structure has stray currents, its durability is affected more by stray currents than general environmental factors [5].

In recent years, with regard to the research on the durability of reinforced concrete under the action of water-saturated environments and stray currents, some studies have found that when the concrete structures working under water-saturated environments are corroded by stray currents, the reinforcement in the concrete will accelerate corrosion and the concrete itself will undergo electrochemical corrosion [6,7]. Ekstrom [8] pointed out that the corrosion parameters of reinforced concrete in water-saturated environments are greater than those in soil environments. Aghajani [9] showed that the hydration product Ca(OH)_2_ of concrete was decomposed under the action of stray currents, which deteriorated the pore structure of concrete. The research of Yang [6] shows that the expansion of the corrosion products of reinforcement caused by stray currents leads to the destruction of concrete structures. Dolara [10] pointed out that the concrete structure may be damaged by stray currents because the average potential of the reinforcement exceeds the limit.

However, in stray current environments, the reinforcement corrosion and calcium ion corrosion of concrete structures in water-saturated environments occur at the same time [11]. Most of the current studies do not consider the influence of reinforcement corrosion and calcium ion corrosion on concrete performance, and its performance degradation mechanism is not clear. Therefore, it is imperative to study the influence of stray currents and calcium ion corrosion on concrete performance in water-saturated environments.

In this paper, the influence of reinforcement corrosion and calcium ion corrosion on water-saturated concrete strength under stray currents is studied.

## 2. Experimental

### 2.1. Experiment Material

The cement was P·O 42.5 ordinary Portland cement. Its chemical composition is shown in Table 1. The fine aggregate was medium sand, with fineness modulus of 2.4 and silt content less than 1.7%; the coarse aggregate was pebble with the maximum particle size of 10 mm. There were two kinds of test water, one was tap water used for the preparation of concrete specimens, and the other was deionized water used for electrochemical tests.

### 2.2. Experiment Design

In order to simplify the later calculation, the cube specimens’ dimensions were 100 mm ×100 mm ×100 mm. They were cured under standard conditions for 28 days. The length of the reinforcement was 80 mm. After derusting and weighing, the external conductor was connected. The erection size was 70.7 mm × 10 mm × 1 mm acrylic plate to ensure that the reinforcement was in the center of the concrete after pouring, as shown in Figure 1.

The specific grouping and main parameters are shown in Table 2. There were 25 test pieces in each group. The energization time of every 5 test pieces was the same, and the energization time was set to 30, 45, 60, 80 and 100 h. In addition, three groups of plain concrete specimens without reinforcement were cast as the control group.

Before the test, the test pieces were immersed in deionized water for 270 h, so that the test pieces were nearly water saturated [12,13], and then a resistivity test was conducted on the concrete test pieces to obtain the resistivity of the concrete before being corroded by stray currents. Next, five test blocks from the same group were placed in the electrolytic cell side by side in order. The reinforcement was perpendicular to the bottom of the electrolytic cell, and a 750 mm steel bar was placed on the long side of the electrolytic cell × 150 mm stainless-steel plate with alligator clips for connecting wires. Deionized water was injected into the electrolyzer, and the water surface was intended to be 10 mm higher than the upper surface of the concrete specimen. The test device is shown in Figure 2.

### 2.3. Test Method

During the electrification test, the corrosion current flowing through the specimen and the development of cracks on the concrete surface were measured. When each group of tests reached the preset test energization time, the power supply was turned off, the test block was taken out of the electrolytic cell, a resistivity test and compressive strength test were conducted on the test block, and the calcium ion dissolution amount in the electrolytic cell was measured. The calcium ion dissolution rate was calculated according to Equations (1) and (2). After the compressive strength test was completed, the reinforcement from the concrete was taken out and the corrosion rate of the reinforcement was calculated according to Equation (3).
(1)$$C_{ca^{2+}}=\frac{m}{M}\times 100\%$$
(2)M=0.001×B×62.1%×1000
(3)$$\gamma =\frac{M_{0}-M_{1}}{M_{0}}$$
where $C_{Ca^{2+}}$ is the calcium ion dissolution rate (%); *M* is the initial mass of CaO contained in a single concrete specimen (mg); *m* is the mass of CaO dissolved from a single concrete specimen (mg); *B* is the amount of cement in concrete mix proportion (kg/m^3^); 62.1% is the mass fraction of CaO in cement clinker; γ is the corrosion rate of reinforcement (%); $M_{0}$ is the original mass of reinforcement (g); $M_{1}$ is the mass of reinforcement after corrosion (g).

## 3. Test Results and Analysis

### 3.1. Analysis of Influence Factors of Reinforcement Corrosion and Concrete Corrosion

Through the test of 9 groups of specimens after corrosion, a total of 45 data of rebar corrosion rates and 45 data of calcium ion dissolution rates were obtained, and the change rules of rebar corrosion rate and calcium ion dissolution rate under different strength grades, rebar diameters and energizing voltage with energizing time were obtained (Figure 3 and Figure 4). The abscissa corresponding to the dotted line in Figure 3 represents the time when the earliest crack occurred in the specimen. It can be seen from Figure 3 and Figure 4 that the change laws of the corrosion rate of reinforcement and the calcium ion dissolution rate of each group of test pieces with the energization time are basically similar. During the early stages of energization, the corrosion quality increases nonlinearly, and the corrosion rate increases continuously. With the increase in power over time, the corrosion rate begins to decrease gradually. The dissolution rate of calcium ions decreases with time, and the dissolution rate reflects the decrease in electric field force.

From Figure 3, it can be concluded that:With the same diameter of reinforcement and the same energizing voltage, the smaller the strength grade of concrete, the more serious the corrosion of reinforcement and the smaller the critical corrosion rate at the time of concrete cracking. For example, after 100 h of power on, the corrosion rate of reinforcement in C25 concrete is 2.32%, and the rate in C35 concrete is 1.98%. Compared with the latter, the corrosion rate of reinforcement in the former increases by 17.17%. This is because the corrosion rate of reinforcement is affected by the water–cement ratio of concrete. The water–cement ratio of C25 concrete is 0.52, and the water–cement ratio of C35 concrete is 0.66. The smaller the water–cement ratio, the denser the concrete paste and the greater the concrete resistance, making the current through the reinforcement in the concrete smaller;With the same strength grade and energizing voltage, in the same energizing time, the larger the diameter of the reinforcement in the concrete specimen, the lower the corrosion rate and the later the crack occurs. It can be seen from Figure 4b that when the energizing voltage is 40 V, the corresponding earliest crack time of C35 concrete with reinforcement diameters of 12 mm, 14 mm and 16 mm is 50.2 h, 52.5 h and 53.2 h, respectively. The larger the diameter of the reinforcement, the smaller the corrosion rate when cracking, indicating that under the same corrosion rate, the larger the diameter of the reinforcement, the greater the rust expansion force caused by corrosion;With the same strength grade and reinforcement diameter, under the same energizing time, the greater the energizing voltage, the more serious the corrosion of reinforcement in the test piece. For example, the final corrosion rate of reinforcement under 40 V corrosion is 2.18 times that of 20 V. It can be seen from Figure 4c that the critical corrosion rate of reinforcement when concrete cracks under high voltage is small, because the higher the voltage, the fluffier the corrosion products [14], and the greater the compressive stress on the surrounding concrete, resulting in a reduction in the critical corrosion rate of cracking.

From Figure 4, it can be concluded that:The corrosion degree of low-grade concrete specimens is more serious than that of high-grade concrete specimens. For example, when the reinforcement diameter is 12 mm, the energizing voltage is 40 V and the energizing time is 100 h, the calcium ion content of C25 test piece is 0.561%, the calcium ion content of C30 test piece is 0.464%, and the calcium ion content of C35 test piece is 0.384%. This is because the water–cement ratio of the concrete with small strength grade is large, the porosity in the cement stone matrix is large, and the calcium ion in the concrete is easier to penetrate into the pore solution;At the same strength level and energizing voltage, the different diameters have no obvious effect on the corrosion results. For example, the calcium ion dissolution rates corresponding to the diameters of the three kinds of reinforcement after 100 h of power on are 0.384, 0.387%, and 0.385%, respectively, and the difference from the average is within 0.1%;When the voltage applied to the test piece is different, the dissolution rate of calcium ions is different. In the selected voltage range, the higher the voltage, the higher the dissolution rate of calcium ions. For example, when the voltage is 20 V, the dissolution rate of calcium ions is 0.27%, while the value is 0.374% when the voltage is 30 V, and the corresponding dissolution rate is 0.464% when the voltage is 40 V.

Nonlinear regression fitting was carried out on the 9 groups of data in the above analysis and the regression equations of each parameter on the corrosion rate and calcium ion dissolution rate of reinforcement were obtained, as shown in Equations (4) and (5).
(4)$$\gamma \left(f_{c},d,U,t\right)=0.665+0.00112t-0.00141f_{c}d+0.00059Ut-0.00018f_{c}^{2}$$
(5)$$C_{Ca^{2+}}\left(f_{c},U,t\right)=0.128-0.0083f_{c}+0.004995U+0.002691t+0.000051Ut$$
where *f_c_* is the original compressive strength (MPa); *d* is the diameter of reinforcement (mm); *U* is the corrosion voltage (V); *t* is the power on time (h); *γ* is the corrosion rate of reinforcement (%); $C_{Ca^{2+}}$ is the calcium ion dissolution rate (%). The determination coefficient R^2^ of the model $\gamma \left(f_{c},d,U,t\right)$ is 0.931; the determination coefficient R^2^ of the model $C_{Ca^{2+}}\left(f_{c},U,t\right)$ is 0.950, which indicates that the two regression equations are highly reliable and can accurately predict the change law of the corrosion rate of reinforcement and the calcium ion dissolution rate.

### 3.2. Analysis of the Interaction between Corrosion of Reinforcement and Calcium Ion Corrosion

#### 3.2.1. Influence of Reinforcement Corrosion on Calcium Ion Dissolution

The larger the corrosion current flowing through the concrete, the faster the ion migration and the more calcium ions are dissolved [8]. When the energizing voltage is kept constant, the corrosion current increases slightly at the initial stage of energization and decreases gradually after reaching the peak value. At the same time, there is a certain synchronization among the changes in corrosion current, corrosion rate of reinforcement and calcium ion dissolution rate. The changes in corrosion rate, corrosion current and increase value of calcium ion dissolution rate with time of C30-14-20, C30-14-30 and C30-14-40 test pieces are shown in Figure 5.

It can be seen from Figure 5a,b that the corrosion current will increase at the initial stage and then decrease with the increase in the corrosion rate of the reinforcement. This is because in the early stage of rebar corrosion, the corrosion has little effect on the circuit resistance. At the same time, under the action of the electric field, a stable current transmission path is formed in the concrete, which makes the corrosion current increase continuously. With the increase in the steel corrosion rate, the total resistance in the circuit increases and the corrosion current begins to decrease. When the corrosion current of C30-14 specimen begins to decrease, the corrosion rate of reinforcement is between 0.281% and 0.379%. It can be seen from the above analysis that, under the condition that the energizing voltage in this test remains unchanged, the corrosion current in other stages will decrease with the increase in the corrosion rate of the reinforcement, except the initial stage of rust corrosion. It can be seen from Figure 5b,c that the calcium ion dissolution rate is consistent with the change law of corrosion current. This is because the corrosion current determines the migration speed of ions in concrete.

Under the condition of keeping the electrified voltage constant, the corrosion of reinforcement affects the corrosion indirectly by affecting the corrosion current, and with the development of reinforcement corrosion, the dissolution rate of calcium ions gradually decreases. Therefore, it can be predicted that the corrosion behavior of calcium ions in concrete under stray currents will slow down with the increase in corrosion degree of reinforcement.

#### 3.2.2. Influence of Calcium Ion Dissolution on Reinforcement Corrosion

Three groups of concrete specimens, C30-14-20, C30-14-30 and C30-14-40, before and after the electrical test were tested by an RST Electrochemical Workstation, and the AC impedance spectra of the specimens were obtained, as shown in Figure 6. The change in the calcium ion dissolution rate of C30-14-20, C30-14-30 and C30-14-40 concrete specimens with the specimens is shown in Figure 7.

It can be seen from Figure 6 and Figure 7 that after the concrete is corroded, the calcium ion dissolution rate increases continuously with the change in the electrification time, and the corresponding concrete electrochemical impedance spectrum shows that the point resistance value constantly shifts to the left side of the abscissa, indicating that the concrete resistivity decreases during the electrification process.

In the AC impedance spectrum, the real resistance corresponding to the smallest imaginary impedance is called the volume resistance and is represented in the diagram as the real resistance at the junction of the high and low frequency regions. The volume resistance is equivalent to the DC resistance of the component under test [15]. According to Figure 7, the volume resistance loss rate of three groups of concrete specimens is obtained, as shown in Table 3.

The data of calcium ion dissolution rate and volume resistance loss rate of three groups of concrete specimens are fitted with Exp function to obtain the functional relationship between volume resistance loss rate *D_r_* and calcium ion dissolution rate, as shown in Equation (6). Figure 8 is a comparison of the volume resistance loss rate test value and the predicted value. It can be seen from Figure 8 that the test value is in good agreement with the predicted value of the formula, and there is a good correlation, indicating that the fitting formula can be used to estimate the volume resistance loss rate of concrete, and can provide a basis for the subsequent research on the electrical resistance performance of concrete under different calcium ion dissolution rates.
(6)$$D_{r}=29.67\times e^{\wedge }\left(-\frac{0.45}{C_{Ca^{2+}}+0.069}\right)$$

As far as this experimental device is concerned, the concrete specimen, electrolyte, reinforcement and stainless-steel plate can be regarded as a circuit with series resistance. Although the concrete resistance gradually decreases with the increase in the calcium ion dissolution rate, the corrosion of reinforcement leads to the increase in the interface resistance between reinforcement and concrete. The corrosion current does not increase with the decrease in the concrete resistivity, so it can be seen that the total resistance of the circuit increases during the electrification process, indicating that the decrease in the concrete resistance is smaller than the increase in the reinforcement concrete interface resistance.

To sum up, calcium ion corrosion slows down the increase in the total resistance value of the circuit, thus slowing down the decrease in the corrosion current of the reinforcement, and then indirectly affects the corrosion of the reinforcement by affecting the corrosion current.

### 3.3. Damage Analysis of Concrete Compressive Strength under Stray Current Corrosion

Under the action of stray currents, the corrosion of reinforcement in concrete and the calcium ion corrosion behavior of concrete will damage the compressive strength of concrete. The loss of concrete compressive strength is the result of the joint action of reinforcement corrosion and calcium ion corrosion.

It can be seen from the analysis in Section 2.1 that the smaller the diameter of the reinforcement or the larger the energizing voltage, the smaller the critical corrosion rate and the calcium ion dissolution rate at the time of concrete cracking. Therefore, it can be inferred that when the corrosion rate of the reinforcement and the calcium ion dissolution rate are the same, the damage effect of the two on the concrete is affected by the diameter of the reinforcement and the stray current intensity, and the damage effect is more significant with the decrease in the diameter of the reinforcement or the increase in the stray current intensity.

According to the test data of 13 groups of test pieces, the three-dimensional curved surface relationship between the corrosion rate of reinforcement, calcium ion dissolution rate and concrete damage degree can be drawn, as shown in Figure 9.

It can be seen from Figure 9 that the damage degree of concrete is closely related to the change in corrosion rate of reinforcement and calcium ion dissolution rate. The damage degree of concrete increases with the increase in the corrosion rate of reinforcement and calcium ion dissolution rate. When the energizing time, energizing voltage and reinforcement diameter are the same, the corrosion rate of reinforcement and calcium ion dissolution rate of C25 concrete are larger than those of C30 and C35, and the damage degree of concrete even reaches 0.38. It can be seen that the corrosion of reinforcement and calcium ion dissolution have a significant impact on the compressive strength of concrete. At the same time, the maximum calcium ion dissolution rate in the 13 groups of data is 0.47%. However, from the comparative analysis, it can be seen that the damage effect of reinforcement corrosion on concrete compressive strength is more obvious than that of calcium ion corrosion.

## 4. Conclusions

The influence of reinforcement corrosion and calcium ion corrosion on concrete strength under the action of stray currents was studied by simulating stray current environments and accelerating tests. The following conclusions were reached:The higher the strength grade, the lower the corrosion of the internal reinforcement and its electrochemical corrosion. With the increase in reinforcement diameter, the corrosion rate decreases, but the calcium ion dissolution rate is not affected by reinforcement diameter. The greater the stray current intensity, the greater the corrosion rate of reinforcement and the dissolution rate of calcium ions and the smaller the critical corrosion rate of reinforcement when concrete cracks under strong currents;The corrosion behavior of calcium ions in concrete under the stray current will slow down with the increase in the corrosion degree of reinforcement;Through nonlinear regression fitting of the test data, the regression equations of the original compressive strength, reinforcement diameter, corrosion voltage, electrification time and other parameters on the corrosion rate and calcium ion dissolution rate of reinforcement were obtained;The damage degree of concrete is closely related to the corrosion of reinforcement and the dissolution of calcium ions. The damage effect of rebar corrosion and calcium ion dissolution on concrete strength is affected by stray current intensity and rebar diameter. When the rebar corrosion rate and calcium ion dissolution rate are the same, the smaller the rebar diameter or the larger the stray current intensity, the more significant the damage effect is. Moreover, the damage effect of rebar corrosion on concrete compressive strength is more obvious than that of calcium ion dissolution.

## Figures and Tables

**Figure 1 materials-15-07287-f001:**
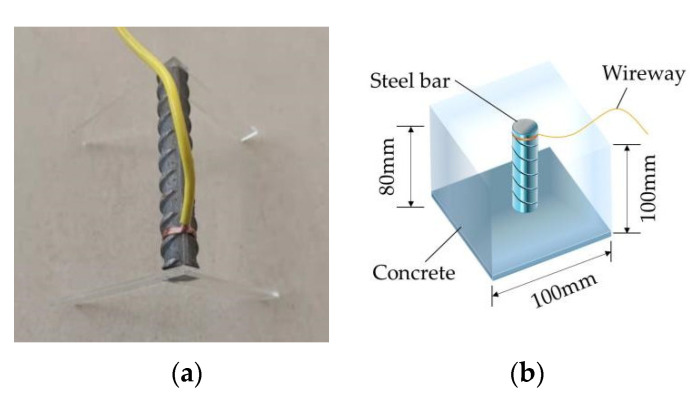
Schematic diagram of erection reinforcement. (**a**) Erection reinforcement; (**b**) Three dimensional schematic diagram of test piece.

**Figure 2 materials-15-07287-f002:**
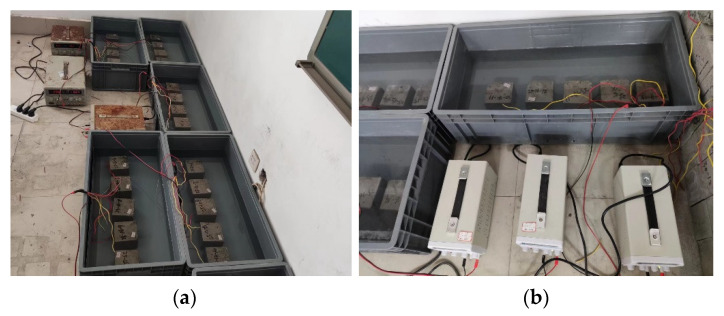
Test device. (**a**) The test piece is immersed in water; (**b**) Corrosion test of steel bar electrification.

**Figure 3 materials-15-07287-f003:**
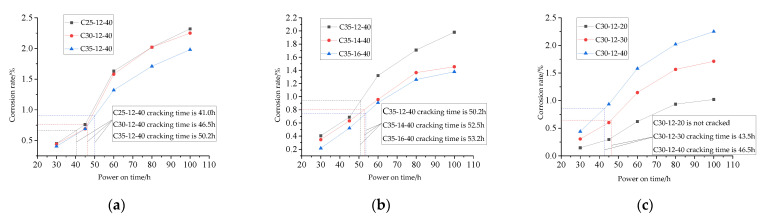
Change curve of corrosion rate of reinforcement. (**a**) Different concrete strength grades; (**b**) Different reinforcement diameters; (**c**) Different energizing voltage.

**Figure 4 materials-15-07287-f004:**
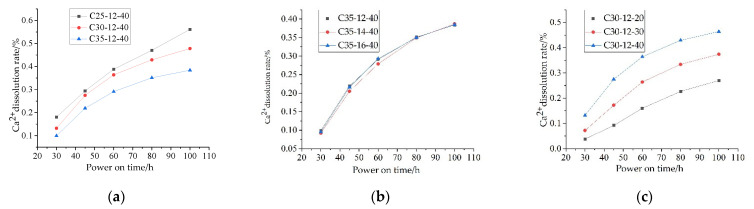
Change curve of calcium ion dissolution rate: (**a**) different concrete strength grades; (**b**) different reinforcement diameters; (**c**) different energizing voltage.

**Figure 5 materials-15-07287-f005:**
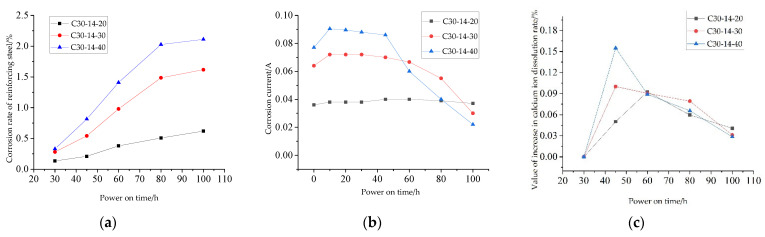
Change curve of corrosion rate of reinforcement. (**a**) Corrosion rate of reinforcement; (**b**) Corrosion current; (**c**) Change value of calcium ion dissolution rate.

**Figure 6 materials-15-07287-f006:**
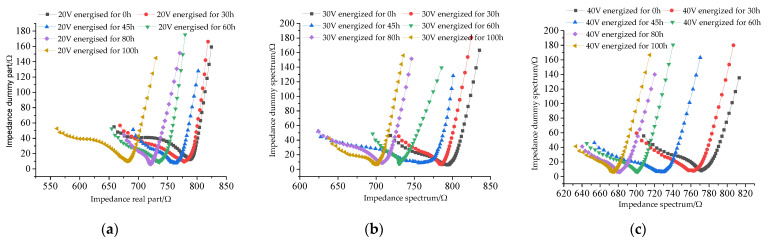
Concrete resistivity. (**a**) C30-14-20 AC impedance spectrum; (**b**) C30-14-30 AC impedance spectrum; (**c**) C30-14-40 AC impedance spectrum.

**Figure 7 materials-15-07287-f007:**
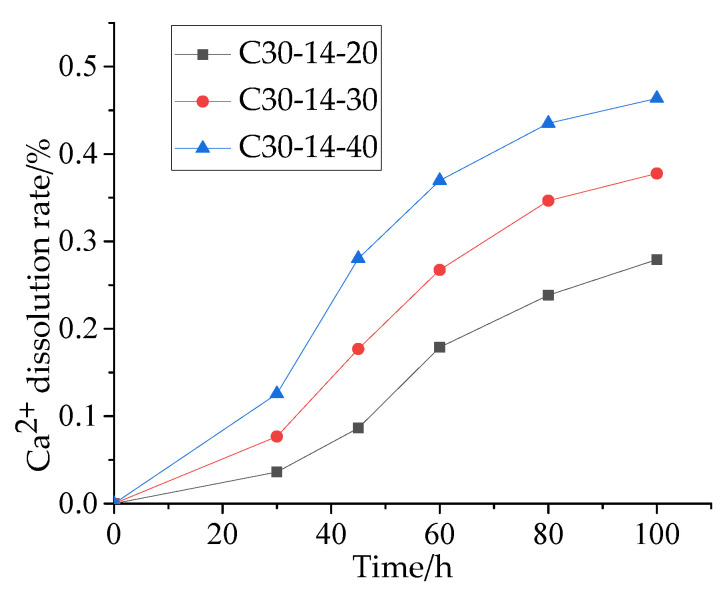
Change of calcium ion dissolution rate.

**Figure 8 materials-15-07287-f008:**
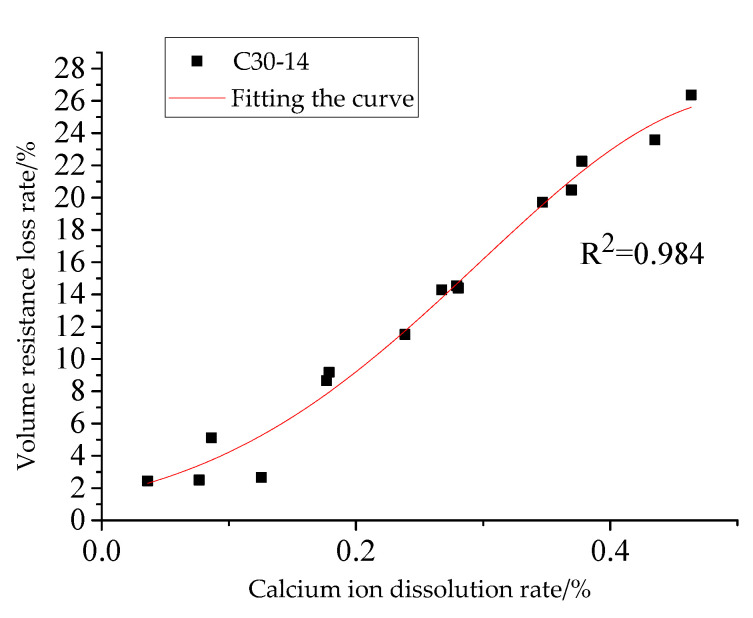
Relationship between calcium ion dissolution rate and volume resistance loss rate.

**Figure 9 materials-15-07287-f009:**
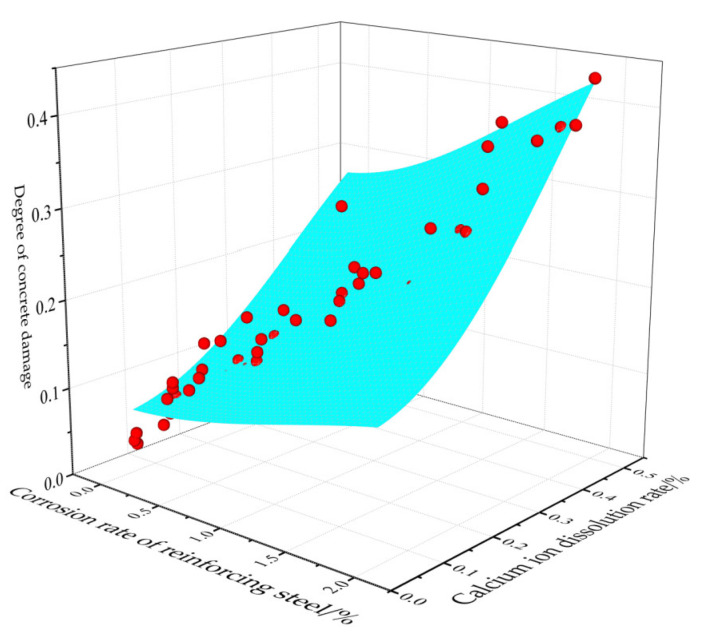
Three-dimensional surface diagram.

**Table 1 materials-15-07287-t001:** Chemical composition of cement (%).

Chemical Composition	CaO	SiO_2_	Al_2_O_3_	SO_3_	FeO	MgO	K_2_O	LOI
P·O 42.5	62.1	21.3	5.7	3.6	2.3	2.0	0.3	2.9

**Table 2 materials-15-07287-t002:** Test grouping and mix proportion.

Group	Strength Grade	Reinforcement Diameter (mm)	Power on Voltage (V)	Water–Cement Ratio	Mix Proportion				
					Sand Ratio (%)	Cement (kg/m^3^)	Sand (kg/m^3^)	Stone (kg/m^3^)	Water (kg/m^3^)
C25-12-40	C25	12	40	0.66	39	303	730	1167	200
C30-12-20	C30	12	20	0.58	36	345	659	1197	200
C30-12-30	C30	12	30	0.58	36	345	659	1197	200
C30-12-40	C30	12	40	0.58	36	345	659	1197	200
C30-14-20	C30	14	20	0.58	36	345	659	1197	200
C30-14-30	C30	14	30	0.58	36	345	659	1197	200
C30-14-40	C30	14	40	0.58	36	345	659	1197	200
C30-16-20	C30	16	20	0.58	36	345	659	1197	200
C30-16-30	C30	16	30	0.58	36	345	659	1197	200
C30-16-40	C30	16	40	0.58	36	345	659	1197	200
C35-12-40	C35	12	40	0.52	33	385	590	1225	200
C35-14-40	C35	14	40	0.52	33	385	590	1225	200

**Table 3 materials-15-07287-t003:** Volume resistance loss rate of concrete under different voltages (%).

Number	Power on Time (h)				
	30	45	60	80	100
C30-14-20	0.66	2.76	6.19	7.99	9.50
C30-14-30	0.93	4.64	8.01	10.77	12.03
C30-14-40	1.18	5.68	9.11	11.60	13.49

## Data Availability

The data are not to be shared.

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
