# Peer review of "Effect of Stray Current on Corrosion and Calcium Ion Corrosion of Concrete Reinforcement"

_materials, 2022, doi:10.3390/ma15207287_

Round 1

Reviewer 1 Report

Comments are listed below:
1. Strengthen the abstract section. Add the key conclusion of the works in the last two lines of the abstract section. Remove the unnecessary information.
2. Discuss the novelty of the work in respect of the application 
3. There are numerous spelling and grammatical errors. Please revise the manuscript thoroughly. Sentences are also not complete and references are also cited in a rough manner. 
4. Try to make a bridge between current and previously published work and specify the gap area and objective of the work. The introduction section is very poor: refer to following published work: 
- Wenchen Ma. (2021). “Behavior of Aged Reinforced Concrete Columns under High Sustained Concentric and Eccentric Loads,” Doctoral Thesis, University of Nevada, Las Vegas, Nevada, USA, 198 p.

5. Provide the image of the experimental setup with good quality. Also, add the image of the welded pipe produced.
6. The results are ok but the discussion section is very poor. It looks like a technical report instead of a technical article. Improve the discussion section and add more references in support of the results.
7. Shorten the length of the conclusion section.
8. The work is good, but the technical discussion and introduction section needs improvement. Paper can be accepted after following minor corrections.

Reviewer 2 Report

Main Issues
English language HAS to be corrected thoroughly. 
It took me more than half of the article to understand that the tests were not performed in "saturated water environment" (From line 56, like they were adding salt to the water in order to check corrosion resistance in different electrolytes, as described as case study in the introduction). In reality the authors were intending water-saturated concrete blocks 

69 - How you were able to assess the precise composition of the concrete? It was a lab-made concrete? You should specify the provenience of every component, or the producer. This is a very important point. In the text you assume the leaching of the sole CaO for the Ca loss calculation.  If other leachable elements are present, they could have an influence on the process.
82 - "nearly water saturated". If it is "nearly", and you don't know the exact amount of this saturation, you are introducing a flaw in your experimental design. Yuo were able to assess this degree of saturation somehow? Please discuss

Tests on concrete are performed after 28 days after its pouring. This time is needed to develop in the concrete the maximum strenght, and consequently to complete the chemical reactions that are responsible for its structure. This is again a very important point in the determination of the quality of the obtained data. You haven't specified the "ageing" of the concrete cubes before the test. 

Too few references added. also, introduction could be improved a lot, for the sake of clarity

229 - You have to describe how you obtained the "Volume resistance loss rate", it has probably also a unit (which is missing at the present state). Whitout this explanation, the 

Minor Issues
7,8 - "Subway, hydropower station and water substation"  Plural
8 - "The influence of stray current on reinforced concrete structure is not clear" partly incorrect
64 - "Rebar shall be construction rebar" both redundant and uncorrect by language

64 to 68 - I think is a copy/paste error. please rewtite/explain/correct147 - v is V
104 - space between reinforcement and parentheses
117 - "At the early stage of power on" please rephrase
149 - "The more fluffy the corrosion products" you should add images

A better description of the EIS workstation and parameters is missing

213 - "The resistivity of concrete determines the electrical resistance of concrete." redundant

Figure 9 - It is difficult to read labels on the x-y plane axes

Round 2

Reviewer 1 Report

You should consider these comments again:

1. Strengthen the abstract section. Add the key conclusion of the works in the last two lines of the abstract section. Remove the unnecessary information.
2. Discuss the novelty of the work in respect of the application 
3. Try to make a bridge between current and previously published work and specify the gap area and objective of the work. The introduction section is very poor: refer to following published work: 
- Wenchen Ma. (2021). “Behavior of Aged Reinforced Concrete Columns under High Sustained Concentric and Eccentric Loads,” Doctoral Thesis, University of Nevada, Las Vegas, Nevada, USA, 198 p.

4. Shorten the length of the conclusion section.

Round 3

Reviewer 1 Report

Accept